# Structural Fluctuation, Relaxation, and Folding of Protein: An Approach Based on the Combined Generalized Langevin and RISM/3D-RISM Theories

**DOI:** 10.3390/molecules28217351

**Published:** 2023-10-30

**Authors:** Fumio Hirata

**Affiliations:** Institute for Molecular Science, National Institute of Natural Sciences, Okazaki 444-8585, Japan; hirata@ims.ac.jp

**Keywords:** generalized langevin theory, RISM/3D-RISM, structural fluctuation, isomerization, protein folding, Gaussian fluctuation, central limiting theorem, solvation free energy, Hessian, harmonic analysis

## Abstract

In 2012, Kim and Hirata derived two generalized Langevin equations (GLEs) for a biomolecule in water, one for the structural fluctuation of the biomolecule and the other for the density fluctuation of water, by projecting all the mechanical variables in phase space onto the two dynamic variables: the structural fluctuation defined by the displacement of atoms from their equilibrium positions, and the solvent density fluctuation. The equation has an expression similar to the classical Langevin equation (CLE) for a harmonic oscillator, possessing terms corresponding to the restoring force proportional to the structural fluctuation, as well as the frictional and random forces. However, there is a distinct difference between the two expressions that touches on the essential physics of the structural fluctuation, that is, the *force constant, or Hessian,* in the restoring force. In the CLE, this is given by the second derivative of the potential energy among atoms in a protein. So, the quadratic nature or the harmonicity is only valid at the *minimum* of the potential surface. On the contrary, the linearity of the restoring force in the GLE originates from the *projection of the water’s degrees of freedom onto the protein’s degrees of freedom*. Taking this into consideration, Kim and Hirata proposed an *ansatz* for the *Hessian matrix*. The ansatz is used to equate the Hessian matrix with the second derivative of the free-energy surface or the potential of the mean force of a protein in water, defined by the sum of the potential energy among atoms in a protein and the solvation free energy. Since the free energy can be calculated from the molecular mechanics and the RISM/3D-RISM theory, one can perform an analysis similar to the normal mode analysis (NMA) just by diagonalizing the Hessian matrix of the free energy. This method is referred to as the Generalized Langevin Mode Analysis (GLMA). This theory may be realized to explore a variety of biophysical processes, including protein folding, spectroscopy, and chemical reactions. The present article is devoted to reviewing the development of this theory, and to providing perspective in exploring life phenomena.

## 1. Introduction

Life phenomena are characterized by a series of chemical reactions and signal transductions [1,2]. The fluctuation and relaxation of proteins from/to the equilibrium structure plays crucial roles in determining the reactivity and the rate of a reaction [3,4,5,6]. For example, in the case of an enzymatic reaction, an enzyme should accommodate substrate molecules first in its cavity or an active site, and then release the product molecules after the chemical reaction to complete a reaction cycle. The protein molecule should significantly change its structure during the two steps of the reaction, molecular recognition and chemical reaction, characterized by Michaelis and Menten [7].

Chemical reactions are characterized by changes in the chemical potential of compounds. The chemical potential of a molecule consists of two parts as clarified by the author in terms of different topics: intramolecular and intermolecular parts [8]. The intramolecular aspect of the chemical potential is determined by its atomic composition and molecular structure, intrinsic to the compound, which are in turn determined by the electronic structure. The intermolecular part of the chemical potential, or “solvation free energy” in the popular terminology, is the aspect that depends on the thermodynamic environment in which the molecule is situated. As has been well documented, those two parts of the chemical potential interplay with each other [9].

The most ubiquitous chemical reaction that takes place in a biological system is the *isomerization reaction*, or the conformational change, which involves a change in the chemical potential of the molecule, but does not cause a change in the atomic composition of the compound [10]. *Protein folding* is one of the most outstanding isomerization reactions, in which changes in thermodynamic conditions such as temperature, pressure, and denaturant concentration cause dramatic changes in the conformation and chemical potential [11]. The structure of a protein before and after the reaction is in a state of thermal equilibrium, and each state fluctuates around its equilibrium conformation in the respective thermodynamic conditions, such as temperature and pressure. One of the most outstanding findings in the fields of biophysics and chemistry, made by Anfinsen, is that protein folding and unfolding are *reversible* upon a change in thermodynamic conditions [11,12,13,14]. This finding strongly suggests that *the structural fluctuation of proteins in each thermodynamic environment is linear or harmonic*. The statement can also be re-phrased as the probability distribution of the fluctuating conformation being *Gaussian* [15].

Experimental evidence for the structural fluctuation being *Gaussian* was provided by Kataoka et al. in a paper published in 1995 [16]. The authors carried out small-angle X-ray scattering (SACS) measurements for myoglobin in water in a variety of thermodynamic conditions, such as native, denatured, and molten globule states, and plotted the logarithm of the scattering intensity against the square of the wave vector, which was represented in a so-called Guinier plot [17] (Figure 1).

The idea of a Guinier plot is based on the following equation for the probability distribution of atoms in a molecule, expressed in Fourier space:$$P(Q)\propto \exp\left[-kQ^{2}\right],$$
where *k* is the inverse of the *correlation length* of the structural fluctuation, or the displacement of atoms from the equilibrium position, $k=1/\langle \Delta R^{2}\rangle$ [16]. All the plots corresponding to the different states of the protein exhibited linear behavior with negative slopes in the small *Q* region; the larger the degrees of denaturation, the greater the slope. The behavior is unequivocal evidence for the distribution of the structural fluctuation in the smaller *Q* region being “*Gaussian*”. Since the small *Q* region of structural fluctuations corresponds to a *collective* motion, the results indicate that the collective fluctuation of the protein is “Gaussian” irrespective of the states of the protein, *native*, *molten globule*, or *denatured*. The collective fluctuation of native conformation has the least *k*, and the same slope extends over a wide range of *Q*. This is because the positional correlation between the fluctuation of atoms, or $\langle \Delta R^{2}\rangle$, of the native state is largest among all the conformational states of the protein. There is another interesting feature seen in the plot of the denatured states of the protein, that is, the change in *k* around *Q* ~0.005 cm^−2^. Such larger *k* values correspond to a local region of the structure. Therefore, the change in *k* reflects the transition in the mode from collective motion to a local mode such as oscillatory motion of individual amino-acid residues.

If the structural fluctuation is harmonic as is indicated by Kataoka et al., the protein folding can be viewed as a transition between the distributions of two conformations, which have different variances. Such a picture of protein folding has been proposed by Akasaka and his coworkers by means of high-pressure NMR [18,19,20]. They have observed a continuous change in the chemical shift as pressure increases, which indicates a shift in the peak of the conformational distribution of the protein from native to denatured, as was verified by the structural analysis based on the two-dimensional (2D) NMR.

It is worth remarking that the Gaussian or harmonic behavior of the structural fluctuation of the protein will not be observed, if the protein molecule is placed in *vacuum*. This is because the interactions among atoms in a protein by themselves are never harmonic, unless the protein is cooled down to a potential minimum, as has been well documented in studies based on normal mode analysis (NMA) [21,22,23,24,25,26]. It is the degrees of freedom of water molecules that make the structural fluctuation of a protein *harmonic*. Since the degrees of freedom of water molecules are essentially *infinity*, their interaction with a protein molecule makes the structural fluctuation of the protein to be Gaussian due to the *central limiting theorem* [27,28,29].

The theorem says that the probability distribution of randomly fluctuating variables around its average value becomes Gaussian, or the normal distribution, unless some of the fluctuations are extraordinarily large. The so-called freely jointed model for the distribution of end-to-end distance (ETED) of a polymer may be a good example for explaining the theorem [30]. In the model, a polymer is expressed by a freely jointed chain of segments, in which the length of each segment is fixed, but the bending and torsion angles among segments are freely varied (Figure 2). The physics of such a model polymer can be readily mapped onto the random walk model of Brownian motion to give a Gaussian distribution for the ETED at the limit *N* >> 1, where *N* is the number of segments making the polymer, that is,
$$w(R)\propto \exp\left(-\sigma^{2}R^{2}\right).$$

In the equation, *R* is the ETED, and σ defines the variance of the distribution as
$$\sigma ={\left(\frac{3}{2N\Delta R^{2}}\right)}^{1/2},$$
where ΔR denotes the fluctuation in a segment of the polymer (Figure 2). The important point to be made is that the distribution of the ETED becomes *Gaussian* only at the condition *N* >> 1. It is worth noting that the Fourier transform of the Gaussian distribution will give the linear behavior in the Guinier plot, as is shown in Figure 1.

In the case of a protein in *vacuum*, the number of variables is not so large, ~10^4^, for the central limiting theorem to be satisfied. On the other hand, a protein in water has essentially an infinite number of degrees of freedom, due to water molecules, the number of which is ~10^23^/mol. So, the structural fluctuation of the protein in water is dominated by the overwhelmingly large degrees of freedom of the solvent.

Theoretical proof for the structural fluctuation of a protein to be *Gaussian* was given by Kim and Hirata in 2012 based on the generalized Langevin theory of a protein in water [31,32,33]. Starting from the Liouville equation for a system consisting of a protein and water molecules, they projected all the degrees of freedom in phase space onto the four dynamic variables: atomic coordinates of a protein and their momentum, the density field of atoms of water, and the flux. The projection essentially produced two equations, one for the solute coordinates and the other for the density field of the solvent, both of which have the same architecture with the classical Langevin equation for the dumped harmonic oscillator in a viscous fluid [34,35]. In the case of a solute or protein, the force acting on solute atoms consists of three contributions, the restoring force proportional to the displacement of positions from equilibrium, the friction due to solvent, and the random force originated from the thermal motion. Since the restoring force on atoms is proportional to their displacement, the energy or free-energy surface in this case is identified as *harmonic*. So, the structural dynamics of a protein has been dramatically simplified to a *harmonic oscillator* immersed in a viscous fluid, excited by thermal motion and dumped by the friction, both originating from solvent [31].

The most important question for solving the structural dynamics of a protein is what the restoring force is, and how the *force constant* or *Hessian* can be formulated. Kim and Hirata have proposed an *ansatz* in which the force constant is identified as the second derivative of the free-energy surface, consisting of the intramolecular potential energy (*U*) of a protein and the solvation free energy (Δ*μ*), with respect to the atomic coordinate of the protein [31]. The ansatz makes the realization of the harmonic analysis of a biomolecule in water feasible, because the second derivatives of both *U* and Δ*μ* can be analytically calculated as functions of the atomic coordinates of the biomolecule, *U* by means of molecular mechanics, and Δ*μ* through RISM/3D-RISM theory. We refer such a harmonic analysis based on generalized Langevin theory and RISM/3D-RISM theory to *Generalized Langevin Mode Analysis* (GLMA).

The present paper is devoted to reviewing the theoretical studies concerning the structural fluctuation of a protein in water, carried out by Hirata and his coworkers since 2012.

## 2. Brief Review of the Kim–Hirata Theory

In the present section, we briefly review the Kim-Hirata theory to analyze the structural fluctuation of a protein in water [31].

The Kim-Hirata theory begins with the Liouville equation that describes the time evolution of dynamic variables **A**(*t*) in phase space:(1)$$\frac{dA(t)}{dt}=i\hat{L}A(t).$$

In the equation, $\hat{L}$ is the Liouville operator to drive the time evolution of the vector **A**(*t*), so-called *dynamic variables,* defined by
(2)$$A(t)\equiv \left(\begin{array}{c} \Delta R_{\alpha }(t)\\ P_{\alpha }(t)\\ \delta \rho_{a}(r,t)\\ J_{a}(r,t) \end{array}\right).$$
where the Greek subscript α and the Roman subscript *a* denote atoms in protein and solvent molecules, respectively. The variables $\Delta R_{\alpha }(t)$ and $P_{\alpha }(t)$ represent the structural fluctuation of the protein, and its conjugate momentum, respectively, while $\delta \rho_{a}(r,t)$ and $J_{a}(r,t)$ are the density fluctuation of the solvent around the protein, and its momentum or the flux, defined by
(3)$$\Delta R_{\alpha }(t)\equiv R_{\alpha }(t)-\langle R_{\alpha }\rangle ,\;P_{\alpha }(t)\equiv M_{\alpha }\frac{d\Delta R_{\alpha }}{dt},$$
(4)$$\delta \rho_{a}(r,t)\equiv \sum_{i}\delta (r-r_{i}^{a}(t))-\langle \rho_{a}\rangle ,\;J_{a}(r,t)\equiv \sum_{i}p_{i}^{a}\delta (r-r_{i}^{a}),$$
where $\langle \cdots \rangle$ denotes an ensemble average of the variables.

The Liouville operator $i\hat{L}$ is defined by the Hamiltonian *H* of the system consisting of a protein molecule in water, that is,
(5)$$H=H_{0}+H_{1}+H_{2}$$
(6)$$H_{0}=\sum_{i=1}^{N}\sum_{a=1}^{n}\left[\frac{p_{i}^{a}\cdot p_{i}^{a}}{2m_{a}}+\sum_{j\neq i}\sum_{b\neq a}U_{0}\left(\left|r_{i}^{a}-r_{j}^{b}\right|\right)\right]$$
(7)$$H_{1}=\sum_{\alpha =1}^{N_{u}}\left[\frac{p_{\alpha }\cdot p_{\alpha }}{2M_{\alpha }}+\sum_{\beta \neq \alpha }U_{1}\left(\left|R_{\alpha }-R_{\beta }\right|\right)\right]$$
(8)$$H_{2}=\sum_{\alpha =1}^{N_{u}}\sum_{i=1}^{N}\sum_{a=1}^{n}U_{\mathrm{int}}\left(\left|R_{\alpha }-r_{i}^{a}\right|\right)$$
where *H*_0_, *H*_1_, and *H*_2_ are the Hamiltonian of a solvent, solute, and the interaction between them, respectively. The subscripts *i* and *j* specify molecules in a solvent, *a* and *b* distinguish atoms in a water molecule, and Greek characters α and β label atoms in the biomolecule. The Liouville operator is defined by
(9)$$i\hat{L}\equiv i{\hat{L}}_{0}+i{\hat{L}}_{1}$$
(10)$$i{\hat{L}}_{0}\equiv \sum_{i=1}^{N}\sum_{a=1}^{n}\left[\frac{1}{m_{a}}p_{i}^{a}\cdot \frac{\partial }{\partial r_{i}^{a}}-\sum_{j\neq i}\sum_{b\neq a}\frac{\partial U_{0}(r_{ij}^{ab})}{\partial r_{i}^{a}}\cdot \frac{\partial }{\partial p_{i}^{a}}-\sum_{\alpha =1}^{N_{u}}\frac{\partial U_{\mathrm{int}}\left(\left|R_{\alpha }-r_{i}^{a}\right|\right)}{\partial r_{i}^{a}}\cdot \frac{\partial }{\partial p_{i}^{a}}\right]$$
(11)$$i{\hat{L}}_{1}\equiv \sum_{i=1}^{N}\sum_{a=1}^{n}\left[\frac{1}{M_{\alpha }}P_{\alpha }\cdot \frac{\partial }{\partial R_{\alpha }}+F_{\alpha }\cdot \frac{\partial }{\partial P_{\alpha }}\right]$$
where *U*_0_ is the intermolecular interaction energy among atoms in a solvent, and *U*_int_ is the interaction energy between atoms in the biomolecule and those in solvent.

Following the recipe of generalized Langevin theory (GLT), Kim and Hirata projected all the mechanical variables in phase space onto **A**(*t*), defined by Equation (2), to essentially derive two GLEs for the time evolution of the dynamic variables, one for the dynamics of a biomolecule and the other for that of a solvent [31].

The projection operator $\hat{P}$ operating on a function *f* in phase space is defined as
(12)$$\hat{P}f\equiv (A,f){\left(A,A\right)}^{-1}A,$$
where the bra-ket (**a**,**b**) represents the scalar product of the vectors **a** and **b** in phase space defined as
(13)$$(a,b)\equiv \langle a^{*}b\rangle =\frac{1}{Z}\int a^{*}(\Gamma )b(\Gamma )\exp\left[-H(\Gamma )/k_{B}T\right]d\Gamma$$
in which *H*(Γ) represents the Hamiltonian of the system defined by Equations (5)–(8).

Operating $\hat{P}$ onto Equation (1) produces four equations of motion in the form of the Langevin equation, two for the structural dynamics of a solute molecule and the other two for the fluctuation in the density field of the solvent. Here, we just focus on the ones that are relevant to the structural fluctuation of a solute molecule, which read
(14)$$\frac{d\Delta R_{\alpha }(t)}{dt}=\Delta V_{\alpha }(t)$$
(15)$$\begin{array}{ll} \frac{d\Delta V_{\alpha }(t)}{dt}= & -\frac{k_{B}T}{M_{\alpha }}\sum_{\beta }{\left(L^{-1}\right)}_{\alpha \beta }\cdot \Delta R_{\alpha }(t)\\ & -\int_{0}^{t}ds\sum_{\beta }\frac{1}{M_{\alpha }}\Gamma_{\alpha \beta }(t-s)\cdot \Delta V_{\beta }(s)+W_{\alpha }(t) \end{array}$$

The equations can be put together into a single equation as
(16)$$M_{\alpha }\frac{d^{2}\Delta R_{\alpha }(t)}{dt^{2}}=-\sum_{\beta }A_{\alpha \beta }\Delta R_{\beta }(t)-\int_{0}^{t}ds\sum_{}\Gamma_{\alpha \beta }(t-s)\cdot \frac{d\Delta R_{\alpha }(s)}{ds}+W_{\alpha }(t),$$
where the second and third terms on the right-hand side represent the frictional force exerted by the solvent and the random force due to thermal excitation, which are related to each other through the fluctuation dissipation theorem [28] (here, details of the expressions concerning the two terms are entirely skipped). The physical meaning of the equation is as follows: a structural fluctuation is excited by thermal motion, expressed by $W_{\alpha }(t)$, and the equilibrium structure is restored by the force in the first term. The second term just represents the frictional force proportional to the relaxation rate of the fluctuation. It is the first term we focus on in the present paper, which looks like that of a harmonic oscillator: the *restoring force* is proportional to the displacement of atoms from their equilibrium positions, or to the *structural fluctuation*. In that respect, the equation is *formally* equivalent to that of a *dumped harmonic oscillator* in a viscous fluid. By ignoring the second and third terms of Equation (16), one finds an equation analogous to stationary dynamics of a harmonic oscillator:(17)$$M_{\alpha }\frac{d^{2}\Delta R_{\alpha }(t)}{dt^{2}}=-\sum_{\beta }A_{\alpha \beta }\Delta R_{\beta }(t).$$

In the equation, the *characteristic or intrinsic frequency* $A_{\alpha \beta }$ is related to the (α,β)-element of the inverse of matrix **L** by
(18)$$A_{\alpha \beta }=k_{B}T{\left(L^{-1}\right)}_{\alpha \beta }$$
where **L** is the variance–covariance matrix of the structural fluctuation of the biomolecule, defined as
(19)$$L\equiv \langle \Delta R\Delta R\rangle .$$

The form of Equation (17) indicates that the energy surface to originate the restoring force is *quadratic* in the displacement vector or fluctuation, and the probability distribution of the fluctuation is *Gaussian,* the variance–covariance matrix of which is **L** defined by Equation (19).

At this point, some readers may raise a question. Why can the free-energy surface of the protein in water possibly be quadratic? Of course, the potential-energy surface of the protein itself is *never* quadratic. As is seen in any computer program of the molecular-dynamics simulation, the interactions among atoms in the protein as well as those with water molecules involve non-harmonic interactions, including the Lennard–Jones as well as the electrostatic interactions. For such systems, the potential-energy surface becomes strictly harmonic only when the system is cooled down to the *global minimum*. That is the essential requirement for the normal mode analysis (NMA) carried out earlier by several authors [21,22]. Then, how is the probability distribution of the structural fluctuation possibly Gaussian?

A quick answer to the question is the *central limiting theorem* [27,28,29]. It is worth remark again that the harmonicity expressed by Equation (17) theoretically proves the experimental finding made by the small-angle X-ray scattering (SAXS) of a protein in water, introduced in the Introduction section of the present paper (Figure 1).

It is standard procedure in theoretical physics and chemistry to express the potential of the mean force, or the *free energy*, of a biomolecule in water as a sum of the interaction energy $U\left(\left\{R\right\}\right)$ among atoms in a protein and the *solvation free energy* $\Delta \mu \left(\left\{R\right\}\right)$. That is,
(20)$$F\left(\left\{R\right\}\right)=U\left(\left\{R\right\}\right)+\Delta \mu \left(\left\{R\right\}\right),$$
where $\left\{R\right\}\equiv \left(R_{1},R_{2},\cdots ,R_{\alpha },\cdots ,R_{N}\right)$ represents a set of coordinates of atoms in a biomolecule [36]. As is implied by Equation (20), Δ*μ* is an implicit function of the solvent coordinates $\left\{R\right\}$, the degrees of which are *projected onto* the solute. Due to the projection, the probability distribution of the structural fluctuation becomes Gaussian as
(21)$$w_{conf}\left(\left\{\Delta R\right\}\right)=\sqrt{\frac{A}{{\left(2\pi \right)}^{3N}}}\exp\left[-\frac{1}{2}\sum_{\alpha }\sum_{\beta }A_{\alpha \beta }\Delta R_{\alpha }\Delta R_{\beta }\right]$$

Based on the theoretical conclusion, Kim and Hirata proposed an *ansatz* that plays a crucial role for further developing the theory [31]. The ansatz equates the *force constant* $A_{\alpha \beta }$ of the restoring force acting on protein atoms, or the inverse of the variance–covariance matrix, to the second derivative of the free-energy surface of a protein molecule in solution. That is,
(22)$$A_{\alpha \beta }=\frac{\partial^{2}F\left(\left\{R\right\}\right)}{\partial \Delta R_{\alpha }\partial \Delta R_{\beta }}.$$

Since it is possible to calculate the solvation free energy $F\left(\left\{\Delta R\right\}\right)$ by means of RISM/3D-RISM theory, the ansatz makes the calculation of the force constant in *solution* feasible. The ansatz has a mathematical isomorphism with the ordinary *force constant* $k_{\alpha \beta }$ in the *harmonic oscillator*, which is defined by
(23)$$k_{\alpha \beta }=\frac{\partial^{2}U\left(\left\{R\right\}\right)}{\partial \Delta R_{\alpha }\partial \Delta R_{\beta }},$$
where $U\left(\left\{R\right\}\right)$ is the interaction potential energy among atoms in the molecule.

## 3. Realization of the Structural Fluctuation of Biomolecules in Solution: Generalized Langevin Mode Analysis

In the preceding section, the author shows, based on generalized Langevin theory, that the structural fluctuation of a biomolecule in water is strictly *harmonic*, and the probability distribution is *Gaussian*. Then, the rational step to be followed is a *harmonic analysis*, similar to the *normal mode analysis*, to the system fluctuating on the free-energy surface defined by Equation (20), that consists of the two terms, the potential energy $U\left(\left\{R\right\}\right)$ among atoms in a biomolecule and the solvation free energy $\Delta \mu \left(\left\{R\right\}\right)$. The harmonic analysis consists of two steps: (i) calculating the Hessian matrix of the free-energy surface with respect to the atomic coordinates of biomolecules, and (ii) solving the eigenvalue problem of the Hessian matrix [37].

### 3.1. Calculation of the Hessian Matrix

The calculation of the second derivative of $U\left(\left\{R\right\}\right)$ has been a routine task for the molecular simulation community [21,22,23,24,25,26]. However, that for the solvation free energy $\Delta \mu \left(\left\{R\right\}\right)$ is a non-trivial problem by any means. First, it is impossible for the molecular simulation, because the molecular simulation does not give the solvation free energy as an explicit function of the atomic coordinates {**R**} of a biomolecule in water. The method may be able to calculate the potential of the mean force projected along a very limited number of the *reaction coordinates* by means of the umbrella sampling, for example, but that is it. It will never be able to calculate the free-energy surface for the entire conformational space spanned by {**R**}. So, it is impossible by any means for the method to calculate the first and second derivatives of the solvation free energy.

On the other hand, the calculation of $\Delta \mu \left(\left\{R\right\}\right)$ as a function of the atomic coordinates $\left\{R\right\}$ of a biomolecule is a routine task for the methods based on the statistical mechanics of liquids represented by RISM/3D-RISM theory [29].

The method to calculate the first derivative, or the “solvent-mediated force”, has been derived by Yoshida and Hirata as
(24)$$\frac{\partial }{\partial R_{\alpha }}\Delta \mu \left(\left\{R\right\}\right)=\sum_{i}\rho_{i}\int \frac{\partial u_{i}^{uv}\left(r;\left\{R\right\}\right)}{\partial R_{\alpha }}g_{i}^{uv}\left(r;\left\{R\right\}\right)dr,$$
where $\rho_{i}$ is the number density of solvent atom *i*, $u_{i}^{uv}\left(r;\left\{R\right\}\right)$ is the interaction between the solute molecule and solvent atom i residing at position **r**, and $g_{i}^{uv}\left(r;\left\{R\right\}\right)$ is the interaction between the solute molecule and solvent atom *i* residing at position **r**, and is the spatial distribution of the solvent atom *i* around the solute [38]. The equation has been first applied by Miyata and Hirata to the molecular-dynamics simulation combined with the RISM/3D-RISM method to calculate the *solvent-induced force* acting on an atom of a solute molecule in water [39]. The method has been implemented in the MD-simulation software, AMBER, by Kovalenko and Omelyan, to further accelerate the simulation of a biomolecule in water [40,41].

In order to calculate the Hessian matrix, or Equation (22), we have to carry out the derivative of Equation (20) with respect to the atomic coordinate of solute, that is,
(25)$$\frac{\partial^{2}}{\partial R_{\alpha }\partial R_{\beta }}\Delta \mu \left(\left\{R\right\}\right)=\sum_{i}\rho_{i}\int \frac{\partial^{2}u_{i}^{uv}\left(r;\left\{R\right\}\right)}{\partial R_{\alpha }\partial R_{\beta }}g_{i}^{uv}\left(r;\left\{R\right\}\right)dr+\sum_{i}\rho_{i}\int \frac{\partial u_{i}^{uv}\left(r;\left\{R\right\}\right)}{\partial R_{\alpha }}\frac{\partial }{\partial R_{\beta }}g_{i}^{uv}\left(r;\left\{R\right\}\right)dr.$$

It is a simple task to calculate the first term, which involves the second derivative of the interaction energy between the solute and solvent with respect to the atomic coordinates of the solute molecule.

On the other hand, it is a non-trivial problem to calculate the second term, because it involves the derivative of the spatial distribution function $g_{i}^{uv}\left(r;\left\{R\right\}\right)$. Fortunately, the recipe to calculate such derivatives of the spatial distribution function has already been proposed by Yu and Karplus a few decades ago [42]. The method is closely related to the numerical solution of the RISM/3D-RISM equation to calculate $g_{i}^{uv}\left(r;\left\{R\right\}\right)$.

Although the method is common to any closure to solve the equation, we just present the procedure corresponding to the Kovalenko–Hirata closure [43]. Then, the RISM/3D-RISM equation consists of the two equations, which are written as
(26)$$\begin{array}{ll} h_{j}^{uv}\left(r;\left\{R\right\}\right) & =\sum_{}\int \chi_{jl}^{vv}\left(\left|r-r^{\prime }\right|;\left\{R\right\}\right)c_{l}^{uv}\left(r^{\prime };\left\{R\right\}\right)dr^{\prime }\\ & \equiv \chi_{jl}^{vv}*c_{l}^{uv}\left(r;\left\{R\right\}\right) \end{array}$$
(27)$$h_{j}^{uv}\left(r;\left\{R\right\}\right)=\left\{\begin{array}{c} \exp\left[d_{j}^{uv}\left(r;\left\{R\right\}\right)\right]\;-1\;\mathrm{for}\;d_{j}^{uv}\left(r;\left\{R\right\}\right)\leq 0\\ d_{j}^{uv}\left(r;\left\{R\right\}\right)\;\mathrm{for}\;d_{j}^{uv}\left(r;\left\{R\right\}\right)>0 \end{array}\right.$$
(28)$$d_{j}^{uv}(r)\;\equiv -u_{j}^{uv}(r)/k_{B}T+h_{j}^{uv}(r)-c_{j}^{uv}(r)$$
where $h_{j}^{uv}(r)\equiv g_{j}^{uv}(r)-1$. If one interprets the equation in terms of non-linear response theory, $\chi_{lj}^{vv}$ is the site–site pair correlation function of a solvent that acts as the susceptibility or response function to the perturbation $c_{j}^{uv}(r)$ from the solute molecule. The derivative of the correlation functions with respect to the atomic coordinate of a protein can be written as
(29)$$\frac{\partial h_{j}^{uv}\left(r;\left\{R\right\}\right)}{\partial R_{\alpha }}=\sum_{j}\chi_{jl}^{vv}*\frac{\partial c_{j}^{uv}\left(r;\left\{R\right\}\right)}{\partial R_{\alpha }}$$
(30)$$\frac{\partial h_{j}^{uv}\left(r;\left\{R\right\}\right)}{\partial R_{\alpha }}=\left\{\begin{array}{c} \begin{array}{l} \left(-\frac{1}{k_{B}T}\frac{\partial u_{j}^{uv}\left(r;\left\{R\right\}\right)}{\partial R_{\alpha }}+\frac{\partial t_{j}^{uv}\left(r;\left\{R\right\}\right)}{\partial R_{\alpha }}\right)\exp\left(-u_{j}^{uv}\left(r;\left\{R\right\}\right)/k_{B}T+t_{j}^{uv}\left(r;\left\{R\right\}\right)\right)\\ \;\;\;\;\;\;\;\;\;\;\;\mathrm{for}\;-u_{j}^{uv}\left(r;\left\{R\right\}\right)/k_{B}T+t_{j}^{uv}\left(r;\left\{R\right\}\right)\leq 0 \end{array}\\ -\frac{1}{k_{B}T}\frac{\partial u_{j}^{uv}\left(r;\left\{R\right\}\right)}{\partial R_{\alpha }}+\frac{\partial t_{j}^{uv}\left(r;\left\{R\right\}\right)}{\partial R_{\alpha }}\;\mathrm{for}\;-u_{j}^{uv}\left(r;\left\{R\right\}\right)/k_{B}T+t_{j}^{uv}\left(r;\left\{R\right\}\right)>0 \end{array}\right.$$

The derivatives can be calculated as one of the variables along the course of iteration to find the solutions for the correlation functions themselves [37].

### 3.2. Diagonalization of the Hessian Matrix

It is a standard task for computational science to diagonalize a matrix to find the eigenvalue and vector. There is no theoretical difficulty left in principle. However, a non-trivial computational problem is left behind, which is the size of the Hessian matrix. If one tries to apply the method to a real protein in water, the number of atoms may become ~10^4^ or more. The number of elements of a Hessian for such a protein amounts to a square of the number of atoms, that is, ~10^8^. So, it may be a challenge even for a peta-scale supercomputer such as “Fugaku”. However, the author believes that it will be a challenge worth making.

In the following, we report a preliminary result for a calculation of the Hessian matrix for a small system, a *dipeptide* in water, in order to demonstrate the feasibility of the methodology *in principle* [37].

## 4. A Generalized Langevin Mode Analysis (GLMA) of Alanine Dipeptide in Water

In the present section, the author reviews the calculation of the low-frequency spectrum of an alanine dipeptide in water, based on the theory just reviewed in the preceding sections. The results are compared with the RIKES spectrum reported by Klaas and his coworkers [37,44].

The observed spectrum is a quantity averaged over the molecules, the conformation of which is fluctuating in time as well as in space. Therefore, we carried out a *molecular-dynamics (MD) simulation* of an alanine dipeptide in water, before performing the harmonic analysis to calculate the spectrum. It should be noted, however, that the MD simulation is not a usual one based on the *Newtonian dynamics* on the *energy* surface, because such dynamics will not produce a trajectory that gives the quadratic surface for the harmonic analysis, as is clarified in the preceding sections. The simulation we carried out is dynamic on the *free-energy* surface described by Equation (20), which surely produces a trajectory that meets the condition for a harmonic analysis.

### 4.1. Molecular Dynamics Simulation of an Alanine Dipeptide on the Free-Energy Surface

Skipping all the technical details of the simulation, the author just shows the trajectory of the alanine dipeptide projected on the two-dimensional surface spanned by the two dihedral angles ($\psi_{1},\varphi_{2}$) shown in Figure 3.

A major peak of the distribution is found around $\psi_{1}=150^{\circ }$ and $\varphi_{2}=-60^{\circ }$, marked by (ii) in Figure 3B, which roughly corresponds to the *trans*–*gauche* conformation. There is also a minor peak of the distribution around $\psi_{1}=150^{\circ }$ and $\varphi_{2}=-150^{\circ }$, which roughly corresponds to the *trans*–*trans* conformation.

There is an interesting feature in the trajectory, which is worth noting. There are a few points between the two major conformations in the space spanned by the two dihedral angles. This surely indicates that those are the transient points between the two local minima stated above. Those points will produce a spectrum in the negative-wavenumber region, because the curvature of the free-energy surface at those regions is negative. However, this is not the entire story. Those points between the two local minima in the dihedral-angle space represent local “modes” or a “higher-frequency mode” in terms of the harmonic analysis. Those modes of oscillation are *localized* around one of the dihedral angles. On the other hand, there are modes of oscillation called the “collective mode”, which extends over the entire molecule. The collective mode with the lowest wavenumber will never produce the spectrum in the negative-frequency region in an equilibrium state. This is because such a mode has a single minimum in the corresponding free-energy surface.

Therefore, the points between the two minima in the dihedral-angle space carry dual physical meanings. One of those is a transient state between the two local minima in the dihedral-angle space. This is the mode that causes the negative frequency in the spectrum. The other is a conformation in the collective mode with the lowest wavenumber. Such a mode will never contribute to the spectrum with a negative frequency, because the curvature of the free-energy well corresponding to such a mode will never be negative.

### 4.2. Spectrum from Multiple Snapshots

Figure 4 shows a histogram representation of the wavenumber spectrum of an alanine dipeptide in water, obtained from the Hessian matrix, $A_{\alpha \beta }$, by diagonalizing the matrix and averaging over 1000 snapshots. The spectrum less than 500 cm^−1^ is depicted, along with the corresponding experimental data (line) obtained by Klaas and his coworkers by means of optical-heterodyne-detected Raman-induced spectroscopy (RIKES) [37,44].

Although there are apparent differences observed between the two results, there is a common feature in the two spectra. The two spectra have four peaks between the wavenumbers 0 and 500 cm^−1^, which are relatively close to each other, that is, ~90 cm^−1^, 250 cm^−1^, 370 cm^−1^, and 450 cm^−1^, in the RIKES spectra, while the peaks are at ~90 cm^−1^, ~240 cm^−1^, ~319 cm^−1^, and ~450 cm^−1^ in the GLMA spectra.

There are marked differences between the two spectra in the following respects: (1) the negative-frequency region observed in the GLMA result, which is absent in the experimental data; (2) the small subpeak around 0 cm^−1^ seen in the RIKES result, which is absent in the GLMA spectra; (3) the large difference in the intensity between the two spectra, especially around the 0 to 100 cm^−1^ region; and (4) the relatively large difference, about 50 cm^−1^, in the peak positions around 319 cm^−1^.

The spectrum at the negative-frequency region in the theoretical result apparently corresponds to the transient points between the dihedral angles in the ϕ−ψ plot depicted in Figure 3. The contribution from such transient states to the spectrum will disappear when the trajectory is long enough to satisfy the Ergodic limit.

The second to fourth differences are caused by the different treatments of the spectrum from water molecules in the inhomogeneous environment around a protein. According to the authors of the experimental paper, the RIKES data were obtained by subtracting the intensity concerning *pure solvent* from the overall spectrum [44]. The procedure indicates that the intensity from water molecules interacting with the solute is not excluded from the spectrum. On the other hand, the contribution from such water molecules is entirely disregarded in the theoretical treatment. Those water molecules interacting with the solute, especially via hydrogen-bonds, are likely to be involved in oscillatory motions in lower-frequency modes. So, it is suggested that the large intensity around 100 cm^−1^ is assigned to the intermolecular oscillatory motion of water molecules interacting with the solute. The suggestion is partially supported by a simulation study carried out for water molecules around a monoatomic solute, Na^+^, K^+^, Ne, A, and Xe, to calculate the wavenumber spectrum of water [45]. All the spectra show large intensities between 0 and 300 cm^−1^, which has been assigned by the author to the librational mode of water molecules.

### 4.3. Contributions to the Spectra from Different Conformations

The large discrepancy in the peak positions between the theoretical and the experimental spectra, ~319 cm^−1^ vs. ~370 cm^−1^, requires a structural analysis of the fluctuational mode. Figure 5 illustrates the structure and the fluctuational mode of the dipeptide corresponding to the peak positions in the GLMA spectrum. In the figures, the direction and amplitude of the fluctuation of each atom are illustrated by thick arrows.

As can be seen, all the modes carry a collective characteristic in the sense that the oscillations extend over the entire molecule. For example, the mode with the lowest frequency, or 87 cm^−1^, looks like a “hinge-bending” motion around the C=O carbonyl bond, since the central carbonyl group and the two terminal groups are oscillating in opposite phases. On the other hand, the mode at 452 cm^−1^ seems to be more localized around the N-terminus group. This may be the reason why the frequency is relatively high.

An interesting behavior is seen in the mode assigned to 319 cm^−1^, in which the carbonyl and the amide nitrogen are oscillating in opposite phase. The oscillation is indicative of a water molecule bridging between the two atoms through a hydrogen bond. In order to clarify if there is such a hydrogen-bonded water bridge or not, the solvation structure of the molecule corresponding to the mode, *k* = 319 cm^−1^, was analyzed.

Figure 6 depicts the radial distribution functions (RDFs) of water molecules around the carbonyl oxygen and the amide nitrogen of the dipeptide. The sharp peak at *r*~1.8 Å in the O(peptide)-H(water) RDF in Figure 6c is a manifestation of the hydrogen bond between the carbonyl oxygen and the water hydrogen. The sharp peak at *r*~2.8 Å in the N(peptide)-O(water) RDF is indicative of a strong electrostatic interaction between the amide nitrogen and the water oxygen. The two sharp peaks in RDFs are strong evidence of the water bridge between the carbonyl oxygen and the amide nitrogen. The situation is illustrated by the cartoon in the bottom right of Figure 6. The analysis suggests that the water bridge through the two strong interactions may be the origin of the fluctuational mode assigned to 319 cm^−1^.

Now, let us clarify the difference in the peak positions at ~319 cm^−1^ and ~370 cm^−1^, respectively, in the theoretical and experimental spectra. As we have already mentioned, the experimental spectrum includes both the contributions from the dipeptide and water molecules, while the theoretical one concerns only the interactions within the dipeptide, which include contributions from the solvent *implicitly*. Therefore, the peak in the theoretical spectrum at ~319 cm^−1^ mainly originates from the H(peptide)-O(peptide) interaction bridged by a water molecule. On the other hand, the peak in the experimental spectrum at ~370 is a composite band consisting of the contributions from the H(peptide)-O(peptide) interaction and the water molecule bridging the two atoms in the peptide. The frequency of the mode of the water molecule, which is not accounted for by the theory, may be higher than that of the H(peptide)-O(peptide) interaction, because the water molecule is connected with the two atoms in the peptide through the two strong interactions.

## 5. Perspective

### 5.1. Conformational Transition of a Biomolecule in Water Viewed as a Chemical Reaction

In the preceding sections, the harmonicity or linearity of the conformational fluctuation of a biomolecule in water is clarified. It will be a rational step to apply the theory to a conformational change of a biomolecule in water. Such a conformational change may be viewed as an *isomerization reaction* between two structures, a reactant and product, each of which is fluctuating around the respective free-energy minimum. It is a popular strategy to apply the linear, or non-linear, response theory to such a transition between two states characterized by different free-energy surfaces. The linear response expression for conformational change due to a perturbation was originally derived by Ikeguchi et al. [46]. The same expression was derived later by Kim and Hirata based on a variational principle as follows [31].

Equation (22) implies that the free-energy surface of a biomolecule around an equilibrium conformation is expressed in the integrated form as
(31)$$F\left(\left\{R\right\}\right)=\frac{1}{2}\sum_{\alpha \beta }\Delta R_{\alpha }A_{\alpha \beta }\Delta R_{\beta }.$$

Let us apply a perturbation to the system:(32)$$F\left(\left\{R\right\}\right)=\frac{1}{2}\sum_{\alpha \beta }\Delta R_{\alpha }A_{\alpha \beta }\Delta R_{\beta }+\sum_{\alpha }\Delta R_{\alpha }\cdot f_{\alpha }$$
where $f_{\alpha }$ is the perturbation acting on the atom α of the molecule. The conformational change induced by the perturbation can be derived by the variational principle:(33)$$\frac{\partial F\left(\left\{R\right\}\right)}{\partial \Delta R_{\beta }}=0,$$
which leads to
(34)$${\langle \Delta R_{\alpha }\rangle }_{1}=\frac{1}{k_{B}T}\sum_{\beta }{\langle \Delta R_{\alpha }\Delta R_{\beta }\rangle }_{0}\cdot f_{\beta }.$$

Equation (34) may be applied to the conformational change in which the change in variance, $\langle \Delta R_{\alpha }\Delta R_{\beta }\rangle$, may be negligible. The theory may not be applied to the case of large conformational change, such as *protein folding*, in which the change in the variance will be significant. Nevertheless, the theory can be extended to such cases through a mathematical idea called the *analytical continuation* as follows.

The first step of the method is to divide the entire process of the reaction into several steps, in each of which the variance–covariance matrix is kept constant.
(35)$${\langle \Delta R_{\alpha }\rangle }_{j+1}=\frac{1}{k_{B}T}\sum_{\beta }{\langle \Delta R_{\alpha }\Delta R_{\beta }\rangle }_{j}\cdot f_{\beta }^{\left(j\right)}$$
where sub- or superscript *j* indicates one such step within which the variance–covariance matrix is constant. The entire change in conformation due to the perturbation may be expressed as
(36)$$\langle R_{\alpha }\rangle ={\langle R_{\alpha }\rangle }_{0}+\sum_{j=1}^{N}{\langle R_{\alpha }\rangle }_{j}$$
where *N* denotes the number of steps, in each of which the variance–covariance matrix is invariant. The total number of steps, *N*, should be carefully chosen depending on the problem on which one is focusing.

Figure 7 schematically illustrates the free-energy change upon a chemical reaction, treated by the linear and non-linear response theories, in which the *x*-axis {R} represents the conformation, and ${\left\{R\right\}}_{eq}^{r}$ and ${\left\{R\right\}}_{eq}^{p}$ denote the equilibrium conformations of the reactant and product, respectively.

As an example, let us go back to the introduction section, in which the Guinier plots of myoglobin in various thermodynamic conditions are depicted. In the figure, the plots for *apo*- and *halo*-myoglobin have similar slopes, indicating that both conformations have similar variance–covariance matrices. In such a case, the linear response theory of Equation (34), or *N* = 1 in Equation (36), may be sufficient to describe the conformational change due to the binding of a ligand. On the other hand, the slope for the denatured state at the small *Q* region is quite different from that for the native conformations. In such a case, the linear response description may no longer be valid, and we have to carry out the non-linear response analysis based on Equations (35) and (36) with an appropriate choice of *N*.

There are two types of perturbations applied to aqueous solutions of a biomolecule to induce a structural transition. One of those is the change in a thermodynamic variable such as pressure, temperature, and concentration of a denaturant including urea and guanidine hydrochloride [11,12,13,14]. The other is a mechanical perturbation caused by a mechanical change in a moiety of the protein, such as photoexcitation of a chromophore or substitution of an amino acid [5,6].

### 5.2. Structural Transition Induced by a Thermodynamic Perturbation

The structural change induced by pressure is formulated by Hirata and Akasaka as
(37)$$\langle \Delta R_{\alpha }\rangle ={\left(k_{B}T_{0}\right)}^{-1}\sum_{\beta }{\langle \Delta R_{\alpha }\Delta R_{\beta }\rangle }_{0}{\left(\frac{\partial \Delta \bar{V}}{\partial R_{\beta }}\right)}_{P,T}P,$$
where *P* and $\Delta \bar{V}$ denote pressure and the partial molar volume of a protein in aqueous solution. The formula may be interpreted as follows [47]. The perturbation of pressure induces the force acting on an atom β in the protein, which is the derivative of thermodynamic work, $P\Delta \bar{V}$, with respect to the coordinate of the atom β. The force is propagated to the atom α through the variance–covariance matrix $\langle \Delta R_{\alpha }\Delta R_{\beta }\rangle$ to change the coordinate of atom α.

The structural change induced by temperature is expressed by
(38)$$\langle \Delta R_{\alpha }\rangle =-{\left(k_{B}T_{0}\right)}^{-1}\sum_{\beta }{\langle \Delta R_{\alpha }\Delta R_{\beta }\rangle }_{0}\left(\frac{d\Delta \bar{S}}{dR_{\beta }}\right)T$$
where *T* is the temperature and $\Delta \bar{S}$ denotes the change in the conformational entropy. The physical meaning of the equation is similar to that of Equation (37), just by replacing P and $\Delta \bar{V}$ by *T* and $\Delta \bar{S}$, respectively. The expression requires an analytical expression for $\Delta \bar{S}$ as a function of the coordinates of a protein. Such an expression has been derived by the author based on RISM/3D-RISM theory [48].

The structural change due to a denaturant such as electrolytes is described by
(39)$$\langle \Delta R_{\alpha }\rangle =-{\left(k_{B}T_{0}\right)}^{-1}\sum_{\beta }\sum_{i}{\langle \Delta R_{\alpha }\Delta R_{\beta }\rangle }_{0}\left(\frac{\partial \mu_{i}}{\partial R_{\beta }}\right)N_{i},$$
where *N_i_* and $\mu_{i}$ represent the concentration and chemical potential of the denaturant, respectively. This expression requires the derivative of the chemical potential of the denaturant species *i* with respect to the coordinate of protein atom β. The calculation is also feasible, because the chemical potential of solution component *i* is *analytically* given by means of RISM/3D-RISM theory [29].

### 5.3. Structural Transition Induced by a Mechanical Perturbation

The present section concerns the theory to describe the structural change of a biomolecule induced by a local conformational change, such as the photo-excitation of a chromophore and the substitution of an amino acid [5,6,49,50,51,52]. Such a process may be characterized by a change in the potential energy *U* in Equation (22) due to the perturbation. In order to formulate the perturbation induced at a local moiety of a protein, the potential energy is decomposed into the three contributions as follows:(40)$$U\left(\left\{R\right\}\right)=U_{r}\left({\left\{R\right\}}_{r}\right)+U_{m}\left({\left\{R\right\}}_{m}\right)+U_{rm}\left({\left\{R\right\}}_{r},{\left\{R\right\}}_{m}\right),$$
where ${\left\{R\right\}}_{m}$ and ${\left\{R\right\}}_{r}$ represent a set of coordinates of atoms in the moiety and that of the reference protein without the moiety, respectively; $U_{m}$ and $U_{r}$ denote the potential energy of the respective portion of the protein; and $U_{rm}$ denotes interactions of atoms between the two portions.

Now, we perform a *thought experiment* in which only the moiety portion of the entire biomolecule is replaced by a new one. The difference in the potential energy before and after the replacement may be written as
(41)$$\Delta U\left(\left\{R\right\}\right)=\Delta U_{m}\left({\left\{R\right\}}_{m}\right)+\Delta U_{rm}\left({\left\{R\right\}}_{r},{\left\{R\right\}}_{m}\right),$$
where $\Delta U_{m}$ and $\Delta U_{rm}$ are the change in potential energy among atoms in the moiety and that between the moiety and the reference protein, respectively.

The expression for the perturbation can be obtained by substituting ΔU into Equation (16) as
(42)$$f_{\beta }=-\frac{\partial \Delta U}{\partial R_{\beta }}=-\frac{\partial \Delta U_{rm}\left({\left\{R\right\}}_{r},{\left\{R\right\}}_{m}\right)}{\partial R_{\beta }}.$$
where $R_{\beta }$ denotes the coordinate of an atom in the reference system, and $\Delta U_{rm}$ is the difference between the interaction energy between atoms in the reference protein and those in the moiety before and after the moiety is modified. It should be noted that the derivative of *U_m_* disappears, because it is irrelevant to the coordinate of atom β in the reference protein.

By substituting Equation (42) into Equation (34), one finds
(43)$${\langle \Delta R_{\alpha }\rangle }_{1}=\frac{1}{k_{B}T}\sum_{\beta }{\langle \Delta R_{\alpha }\Delta R_{\beta }\rangle }_{0}\cdot \left(-\frac{\partial \Delta U_{rm}\left({\left\{R\right\}}_{r},{\left\{R\right\}}_{m}\right)}{\partial R_{\beta }}\right),$$
in which ${\langle \Delta R_{\alpha }\Delta R_{\beta }\rangle }_{0}$ is the variance–covariance matrix of the reference system, that is, the protein without the moiety. The linear response expressions, Equation (43), are interpreted as follows. *The force exerted by atoms in the moiety induces the displacement in atom*
β
*of a protein, which propagates through the variance–covariance matrix*
${\langle \Delta R_{\alpha }\Delta R_{\beta }\rangle }_{0}$
*to cause a global conformational change of the molecule,*
${\langle \Delta R_{\alpha }\rangle }_{1}$.

## 6. Concluding Remarks

The recent development of a theoretical method, referred to as Generalized Langevin Mode Analysis (GLMA), was reviewed. The method combines the two theories in statistical mechanics, or generalized Langevin theory and RISM/3D-RISM theory, to calculate the second derivative, or the Hessian matrix, of the free-energy surface of a biomolecule in solution, which consists of the intramolecular interaction among atoms in the biomolecule and the solvation free energy. It has been shown that the Hessian matrix so calculated can be applied to such processes as the spectral analysis of low-frequency modes of a solute in water and the isomerization reaction of a biomolecule in water, which includes the protein folding.

The harmonic analysis of a biomolecule in solution is rationalized by the generalized Langevin equation derived by Kim and Hirata, which is strictly harmonic due to the central limiting theorem. Experimental evidence for the theorem has been given by Kataoka et al. by means of the small-angle X-ray scattering for proteins in a variety of conformations, such as native, molten-globule, and denatured states. The Guinier plots, or the logarithm of the scattering intensity through X-rays plotted against the wave vector, exhibited linearity at least in the low-wave-vector region, where the collective mode of a protein is considered. The finding is unequivocal evidence of the Gaussian distribution of the structural fluctuation, or of the harmonicity.

It should be noted that the linearity of the fluctuation holds within a single mode belonging to a particular eigenfrequency in GLMA. So, if one focuses on higher-frequency modes, or local modes, of a protein in water, such as a torsional-angle fluctuation concerning few amino-acid residues, a “transition” or an “isomerization reaction” from one conformation to the other may occur through a small perturbation, since the free-energy barrier between two such modes may not be so high. The local conformational change associated with the gating mechanism of an ion channel and the induced fitting mechanism of the ligand binding by an enzyme are typical examples of such an isomerization reaction [3,4,5,6]. In such cases, the conformational change should remain within a *linear-response* regime to restore its original conformation for the next reaction cycle. If this is not the case, the protein will lose its function after the perturbation is removed. Therefore, the *linearity* of a local mode of structural fluctuations is essential for a protein to perform its function.

The mode assigned to the lowest frequency, with which the global structure of a protein is concerned, should remain in the same free-energy well, if the thermodynamic condition is not changed. If it is a native condition, the conformation will fluctuate around the minimum of the free-energy well. The conformational fluctuation is caused by a temporal as well as spatial fluctuation of the thermodynamic variable such as temperature and pressure. Such a structural fluctuation caused by that of thermodynamic fluctuation may cause the entire unfolding of a protein as a rare event, as has been observed by Akasaka and his coworkers by means of pressure NMR [18,19,20]. Nevertheless, such unfolding by the fluctuation in local thermodynamic conditions is temporal, and will restore its native conformation quickly, since such an unfolded protein is placed at an extremely high point on the free-energy valley corresponding to the native conformation. Therefore, the *linearity* of the structural fluctuation concerning the global or collective modes is essential for a protein to ensure the robustness of the native conformation.

The GLMA method developed here can be applied to explore the structural fluctuation of a protein in solution without any further development in the theory, but with the assistance of much greater computational power.

## Figures and Tables

**Figure 1 molecules-28-07351-f001:**
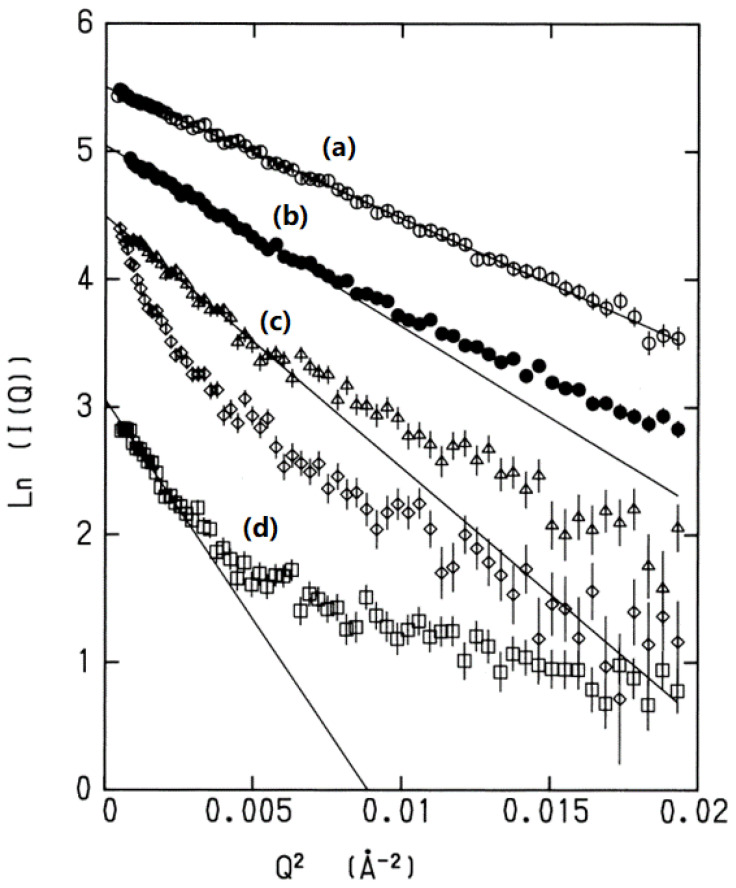
The Guinier plot for a variety of conformations of myoglobin: (**a**) homomyoglobin (native state), (**b**) apomyoglobin (native state), (**c**) molten globule state, and (**d**) unfolded state. Reprinted/adapted with permission from Ref. [16]. More details on “Copyright and Licensing” are available via the following link: https://s100.copyright.com/AppDispatchServlet.

**Figure 2 molecules-28-07351-f002:**
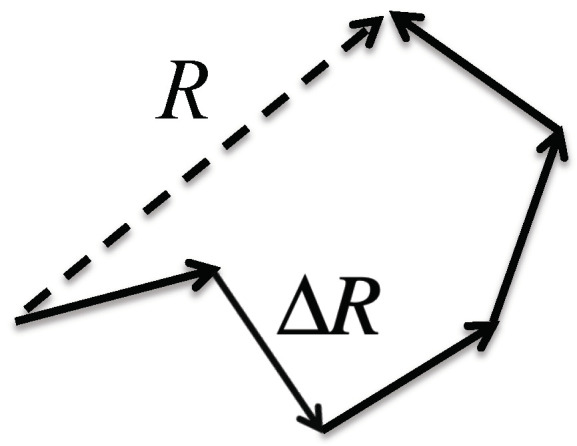
Schematic view to illustrate the freely jointed model of a polymer: *R*, the end-to-end distance; Δ*R*, a segment of polymer.

**Figure 3 molecules-28-07351-f003:**
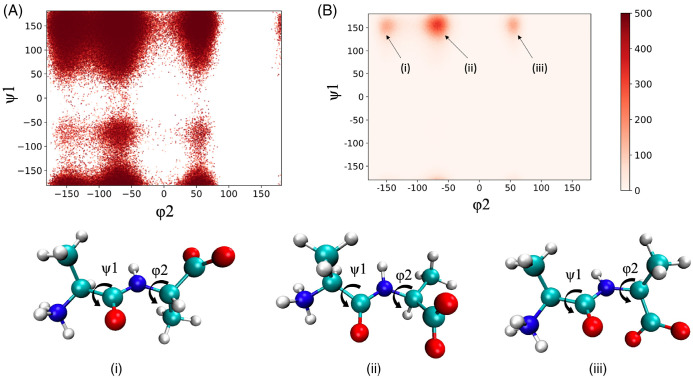
The 3D-RISM/MD trajectory of an alanine dipeptide projected onto the dihedral angle (Ψ1,φ2) space: (**A**) the snapshots at every 80 steps, (**B**) the distribution of the trajectory. The molecular pictures depicted under the two figures (**A**,**B**) illustrate the conformations corresponding to the snapshots (**i**–**iii**) in figure (**B**).

**Figure 4 molecules-28-07351-f004:**
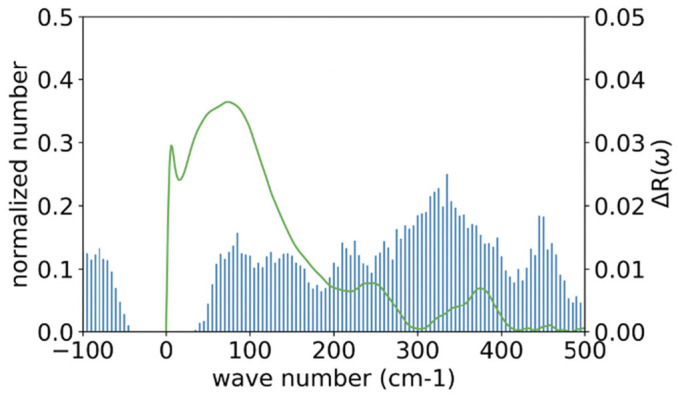
Comparing the low-frequency spectrum from GLMA with that from RIKES: the histogram, the spectrum from GLMA; the green line, the RIKES spectrum (The RIKES data have been provided by Wynne [44]).

**Figure 5 molecules-28-07351-f005:**
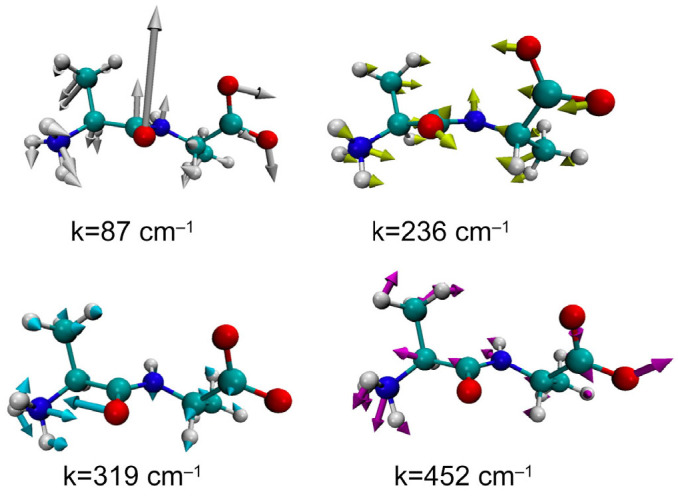
The modes of fluctuation corresponding to the peaks of the wave number spectrum: the thick arrows indicate the direction and magnitude of the oscillatory motion.

**Figure 6 molecules-28-07351-f006:**
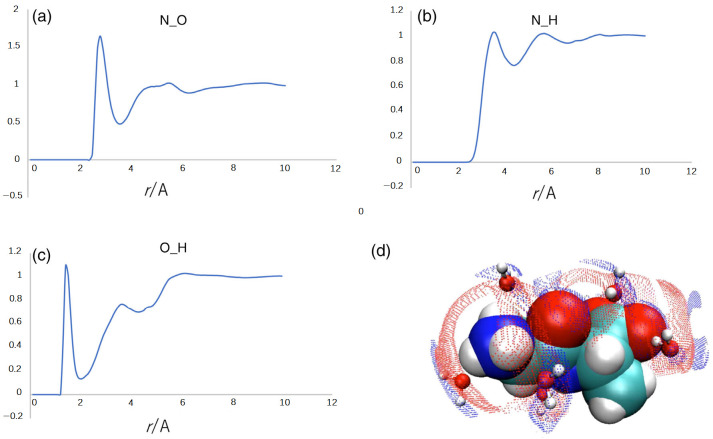
(**a**–**c**) The radial distribution functions (RDFs) of water molecules around the dipeptide in the fluctuational mode with 319 cm^−1^. The figure depicted by the red and blue spots in (**d**) is the 3D distribution of the H and O atoms of water molecules around the dipeptide: red, oxygen; blue, hydrogen. Note that the distribution of the hydrogen atom extends near the carbonyl-oxygen of the peptide, which creates the hydrogen-bond peak at r~1.5 Å in the RDF of the O(peptide)-H(water) pair in (**c**).

**Figure 7 molecules-28-07351-f007:**
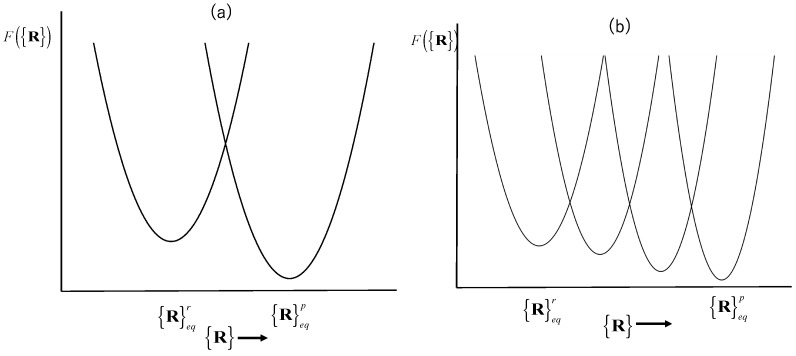
Schematic description of the change in the free-energy surfaces upon chemical reactions: (**a**) the linear response analysis; (**b**) the non-linear response analysis.

## Data Availability

No new data is created.

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
