# Peer review of "Structural Fluctuation, Relaxation, and Folding of Protein: An Approach Based on the Combined Generalized Langevin and RISM/3D-RISM Theories"

_molecules, 2023, doi:10.3390/molecules28217351_

Round 1

Reviewer 1 Report

Th paper is a good review of the author’s theory. A few details detract from the presentation.

1) The abstract would be better served without so many equations so that your work would connect with a wider readership. Just describe what you mean in words. For instance you say “variance- covariance matrix of the fluctuation” but then you define a symbol, with an equation, which is never used in the abstract (later repeated as equation19).  Again, if you say “solvation free energy” there is no need to follow with delta mu.

2) RISM is misspelt in the abstract (RIMS?)

3) Michaelis is misspelt in the introduction

4) In the introduction, many would argue that Anfindsen’s thermodynamic hypothesis of protein folding does not imply  nor is equivalent to” the structural fluctuation of protein in each thermodynamic environment is linear or harmonic”  .

5) Formatting problems in the references where everything has two numbers and sometimes they are not the same. (Like #18)

6) On page 6 the authors start to answer the issue of why they can treat the effective potential harmonically. The explanation as written fails as the author digresses into a central limit derivation argument with out giving the simple physical picture involved. In fact the full argument needs to be stated far before page 6 for clarity. The fact that one can define a quadratic component around a particular conformation or region must be emphasized as well as defining what the region or conformation is.

7)The ansatz given on pg 7 eq 22  seems to be for instantaneous forces or per time step.

8) Figure 3 does not define the lines or symbols. Also the agreement is not good and should be mentioned as such. The fact that one finds 4 totally different peaks in each is not convincing and to say that it may require a structural analysis is a gross understatement.

9) it is hard to reconcile the units in equations 31 - 35 with the perturbation, f.

The manuscript needs English editing.

Author Response

Thank you for the invaluable comments on my paper. 

Let me reply to the comments as follows.

Comment 1) The abstract would be better served without so many equations so that your work would connect with a wider readership. Just describe what you mean in words. For instance you say “variance- covariance matrix of the fluctuation” but then you define a symbol, with an equation, which is never used in the abstract (later repeated as equation19).  Again, if you say “solvation free energy” there is no need to follow with delta mu.

(Reply to the comment)

All the equations in Abstract is removed as follows.

----------------------------

In 2012, Kim and Hirata have derived two generalized Langevin equations (GLE) for a biomolecule in water, one for the structural fluctuation of the biomolecule, and the other for the density fluctuation of water, by projecting all the mechanical variables in the phase space onto the two dynamic variables, the structural fluctuation defined by displacement of atoms from their equilibrium positions, and the solvent-density fluctuation.

   The equation has an expression similar to the classical Langevin equation (CLE) for a harmonic oscillator, having the the terms corresponding to the restoring force proportional to the structural fluctuation, the frictional, and random forces. However, there is a distinct difference between the two expressions, which touches the essential physics of the structural fluctuation, that is the force constant, or Hessian, in the restoring force. In CLE, it is given by the second derivative of the potential energy among atoms in protein. So, the quadratic nature or the harmonicity is only valid at the minimum of the potential surface. On the contrary, the linearity of the restoring force in GLE originates from the projection of the water degrees-of-freedom onto the protein degrees-of-freedom.

   Taking those into consideration, Kim and Hirata proposed an ansatz for the Hessian matrix. The ansatz is to equate the Hessian matrix with the second derivative of the free energy surface or the potential of mean force of a protein in water, defined by a sum of the potential energy among atoms in protein and the solvation free energy. Since the free energy can be calculated, respectively, from the molecular mechanics and the RISM/3D-RISM theory, one can perform an analysis similar to the normal mode analysis (NMA), just by diagonalizing the Hessain matrix of the free energy. The method is referred to as the generalized Langevin mode analysis (GLMA). The theory may be realized to explore a variety of biophysical processes, including protein folding, spectroscopy, and chemical reactions.

   The present article is devoted to review the development of the theory, and to provide the perspective in exploring the life phenomena.

Comment 2) RISM is misspelt in the abstract (RIMS?)

(Author's Reply to the comment)

    Corrected.

Comment 3) Michaelis is misspelt in the introduction

(Author's Reply to the comment)

      Corrected.

Comment 4) In the introduction, many would argue that Anfindsen’s thermodynamic hypothesis of protein folding does not imply  nor is equivalent to” the structural fluctuation of protein in each thermodynamic environment is linear or harmonic”  .

(Author's Reply to the comment)

  The author know certainly  that his view  is not a consensus view concerning Anfinsen's thermodynamic hypothesis. In fact, it is the main reason for writing this paper. So, the sentence concerning the hypothesis is rewritten as follows.

"The finding suggests strongly that the structural fluctuation of protein in each thermodynamic environment is linear or harmonic. The statement can be re-phrased also as the probability distribution of the fluctuating conformation beingGaussian.[15]"         Comment 5) Formatting problems in the references where everything has two numbers and sometimes they are not the same. (Like #18)   (Author's Reply to the comment) The author does not  understand this comment. I don't see any reference which has two numbers.    6) On page 6 the authors start to answer the issue of why they can treat the effective potential harmonically. The explanation as written fails as the author digresses into a central limit derivation argument with out giving the simple physical picture involved. In fact the full argument needs to be stated far before page 6 for clarity. The fact that one can define a quadratic component around a particular conformation or region must be emphasized as well as defining what the region or conformation is.   (Author's Reply to the comment) The author thanks to the reviewer for making this comment. Replying this comment, the author moved the argument to the introduction section.The author added a figure (Fig. I-2) to illustrate  the central limiting theorem.

The theorem says that the probability distribution of randomly fluctuating variables around its average value becomes Gaussian, or the normal distribution, unless some of the fluctuations is extraordinarily large. So-called freely-jointed model for the distribution of end-to-end distance (ETED) of a polymer may be a good example for explaining the theorem.[30] In the model, a polymer is expressed by a freely-jointed chain of segments, in which the length of each segment is fixed, but the bending and torsion angles among segments are freely varied. (Fig. I-2) The physics of such a model polymer can be readily mapped onto the random walk model of the Brownian motion to give a Gaussian distribution for the ETED at the limit N >>1, where N is the number of segments making the polymer, that is,In the equation, R is ETED, and defines the variance of the distribution as,where  denotes the fluctuation of a segment of the polymer. (Fig. I-2) The important point to be made is that the distribution of ETED becomes Gaussian only at the condition, N>>1. It will be worthwhile to be noted that the Fourier transform of the Gaussian distribution will give the linear behavior in the Guinier plot as was shown in Fig. I-1.       

Fig. I-2. Schematic view to illustrate the freely-jointed model of a polymer: R, the end-to-end distance; , a segment of polymer.

 In the case of a protein in vacuum, the number of variables is not so large, ~104, for the central limiting theorem to be satisfied. On the other hand, a protein in water has essentially an infinite number of degrees freedom, due to water molecules, the number of which is ~1023/mol. So, the structural fluctuation of protein in water is dominated by the overwhelmingly large degrees-of-freedom of the solvent.

Comment 7)The ansatz given on pg 7 eq 22  seems to be for instantaneous forces or per time step.

(Author's Reply to the comment)

      Yes, it is. 

Comment 8) Figure 3 does not define the lines or symbols. Also the agreement is not good and should be mentioned as such. The fact that one finds 4 totally different peaks in each is not convincing and to say that it may require a structural analysis is a gross understatement.

(Author's Reply to the comment)

The figure caption is revised as follows in order to explain the lines.

"Fig. IV-2. Comparing the low frequency spectrum from GLMA with that from RIKES: the histogram, the spectrum from GLMA; the green line, RIKES spectrum. (The RIKES data has been provided by K. Wynne.[Ref. 44])"

The disagreement is caused essentially by the difference  in treatment of the contribution from water molecules interacting with protein. It was explained in the paper as follows. 

"The second to fourth differences are caused by the different treatments of the spectrum from water molecules in the inhomogeneous environment around protein. According to the authors of the experimental paper, the RIKES data was obtained by subtracting the intensity concerning pure solvent from the overall spectrum.[44] The procedure indicates that the intensity from water molecules interacting with the solute is not excluded from the spectrum. On the other hand, the contribution from such water molecules are entirely disregarded in the theoretical treatment. Those water molecules interacting with the solute, especially via hydrogen-bond, are likely to be involved in oscillatory motions in lower frequency modes. So, it is suggested that the large intensity around 100 cm-1 is assigned to the intermolecular oscillatory motion of water molecules interacting with the solute. The suggestion is supported partially by a simulation study carried out for water molecules around a monoatomic solute, Na+, K+, Ne, A, and Xe, to calculate the wavenumber spectrum of water.[45] All the spectra show large intensity between 0 to 300 cm-1, which has been assigned by the author to the librational mode of water molecules."

Comment 9) it is hard to reconcile the units in equations 31 - 35 with the perturbation, f.

(Author's Reply to the comment)

The author does not see any inconsistency in the units in 31-35, since the perturbation force, f, times displacement vector has unit of energy.

Reviewer 2 Report

The manuscript is devoted to an extension of the generalized Langevin equations developed by Kim & Hirata.

The paper focuses mainly on an explanation of Gaussian nature of conformational fluctuations arisen for solvated proteins.

The main improvements of the authors are an explanation the gap between the experiments (RIKES data) and theoretical predictions of the authors. The author arguments are well documented and involve a lof of details to prove the theory. There is the section Perspectives which outline the most important impacts of the theory to transitions of solvated proteins. An origin of deviations from the Gassian statistics of fluctuations are briefly discussed. The topic of manuscript is rather original and related to Protein Folding phenomena.

All the references are appropriate and conclusions are consistent with the author arguments. The manuscript is to be published since it seems to be rather interesting for readers of your journal.

One minor remark.

Figure captions for Figs. 1 and 3 are to be improved.

The original paper is to be cited in the case of Fig.1

The original paper is to be cited for the RIKES data and the curves are to be indicated in the case of Fig. 3.

Author Response

The author thank to the referee for evaluating highly for the essential point of the paper.

(Comment )

Figure captions for Figs. 1 and 3 are to be improved.

The original paper is to be cited for the RIKES data and the curves are to be indicated in the case of Fig. 3.

(Reply to the comment)

The figure captions are revised as follows.

Fig. I-1. The Guinier plot for a variety of conformations of myoglobin: (a) homomyoglobin(native state), (b)apomyoglobin(native state), (c) molten globule state, (d) unfolded state. (The figure was reproduced from the paper by Kataoka et. al., [Ref. 16])

Fig. IV-2. Comparing the low frequency spectrum from GLMA with that from RIKES: the histogram, the spectrum from GLMA; the green line, RIKES spectrum. (The RIKES data has been provided by K. Wynne.[Ref. 44])